# Impacts of Air Velocity Treatments under Summer Condition: Part I—Heavy Broiler’s Surface Temperature Response

**DOI:** 10.3390/ani12030328

**Published:** 2022-01-29

**Authors:** Suraiya Akter, Bin Cheng, Derek West, Yingying Liu, Yan Qian, Xiuguo Zou, John Classen, Hernan Cordova, Edgar Oviedo, Lingjuan Wang-Li

**Affiliations:** 1Department of Biological and Agricultural Engineering, North Carolina State University, Raleigh, NC 27695, USA; sakter@ncsu.edu (S.A.); chengbin0228@gmail.com (B.C.); drwest@ncsu.edu (D.W.); lyy@njau.edu.cn (Y.L.); qianyan@njau.edu.cn (Y.Q.); xiuguozou@gmail.com (X.Z.); classen@ncsu.edu (J.C.); 2Department of Electrical Engineering, College of Engineering, Nanjing Agricultural University, Nanjing 210095, China; 3Prestage Poultry Science Department, North Carolina State University, Raleigh, NC 27695, USA; hacordovanoboa@gmail.com (H.C.); eooviedo@ncsu.edu (E.O.)

**Keywords:** heavy broiler, heat stress, surface temperature, air velocity, temperature humidity index

## Abstract

**Simple Summary:**

The surface temperature variation of heavy broilers (42–61 d age) under heat stress is an important indicator of thermal comfort, but it is not well studied and reported yet. This study examined the variation of surface temperatures of broilers through two dynamic air velocity treatments under hot summer conditions. It was discovered that the surface temperatures varied over age, daytime, and environmental factors (air temperature, relative humidity, and temperature humidity index). A simple linear regression model to predict the surface temperature of heavy broilers was developed. The findings from this study will enhance knowledge to understand the broilers’ responses under heat stress, which will be helpful in providing necessary management decisions to create a comfortable thermal environment.

**Abstract:**

Heavy broilers exposed to hot summer conditions experience fluctuations in surface temperatures due to heat stress, which leads to decreased performance. Maintaining a bird’s homeostasis depends on several environmental factors (temperature, relative humidity, and air velocity). It is important to understand the responses of birds to environmental factors and the amount of heat loss to the surrounding environment to create thermal comfort for the heavy broilers for improved performances and welfare. This study investigates the variation in surface temperatures of heavy broilers under high and low air velocity treatments. Daytime, age and bird location’s effect on the surface temperature variation was also examined. The experiment was carried out in the poultry engineering laboratory of North Carolina State University during summers of 2017, 2018, and 2019 as a part of a comprehensive study on the effectiveness of wind chill application to mitigate heat stress on heavy broilers. This live broiler heat stress experiment was conducted under two dynamic air velocity treatments (high and low) with three chambers per treatment and 44 birds per chamber. Surface temperatures of the birds were recorded periodically through the experimental treatment cycles (flocks, 35–61 d) with infrared thermography in the morning, noon, evening, and nighttime. The overall mean surface temperature of the broilers under two treatments was found to be 35.89 ± 2.37 °C. The variation in surface temperature happened due to air temperature, thermal index, air velocity, bird’s age, daytime, and position of birds inside the experimental chambers. The surface temperatures were found lower under high air velocity treatment and higher under low air velocity treatment. During the afternoon time, the broilers’ surface temperatures were higher than other times of the day. It was also found that the birds’ surface temperature increased with age and temperature humidity indices. Based upon the experimental data of five flocks, a simple linear regression model was developed to predict surface temperature from the birds’ age, thermal indices, and air velocity. It will help assess heavy broilers’ thermal comfort under heat stress, which is essential to provide a comfortable environment for them.

## 1. Introduction

As homoeothermic birds, broiler chickens maintain their body temperatures between 40.6–41.7 °C by dissipating heat produced from metabolism [1]. When the environmental temperature goes above a bird’s thermoneutral zone, it becomes challenging for the bird to maintain its thermoregulatory status as it cannot release excess heat. As a result, heat stress occurs, which affects the chicken’s behavioral, physiological and immunological properties [2]. Increased panting [3], mortality rate [4,5] feed conversion ratio [6,7], and decreased feed intake [7,8,9] and body weight gain [6,7,9] are some of the significant effects of heat stress on broiler. Moreover, heat stress hinders overall poultry and egg quality [10] leading to annual loss [11]. Most studies that investigating the broilers’ performance or responses under heat stress conditions were conducted for 1–42-day-old birds. Producers are moving towards bigger broiler production with a market age of 63 d and a bodyweight of 3.75 kg to meet the continuous demand for poultry meat [12,13,14]. However, they are challenged to maintain these heavy birds’ constant performance and welfare due to heat stress, especially in hot and humid summer seasons. Moreover, the existing production structures cannot provide adequate thermal comfort due to the increased incidence and severity of heatwaves in the broiler-producing region caused by climate change. To provide a comfortable thermal environment to the heavy broilers (i.e., 42–61 d), the assessment of the birds’ response to heat stress is of utmost importance.

Some researchers suggested deep body temperature (DBT) as an indicator of stress as it is very responsive to different stress levels [15,16]. Although Hamrita [17] found DBT responses of birds of age 8.6 to 9.4 weeks under heat stress, the method of implementing biotelemetry to obtain DBT was exhaustive and stressful for the birds. Cangar [18] and Giloh [19] found that surface temperatures (Ts) are indicative of the broilers’ comfort or thermal stress. The birds’ Ts vary at different body parts and change with age [18]. Moreover, research has indicated that DBT has a strong correlation with Ts [19]. Besides physiology, the environment also impacts Ts [18,19,20]. Studies showed the Ts changes with air temperature (Ta) and relative humidity (RH). An increase in Ta and RH increases the Ts of broilers of age 1–35 d [21,22,23]. The Ts variation with all these environmental factors was assessed for the birds of age up to 42 d. Hence, although Ts can be used as an indicator of stress, the evaluations are not available for heavier and bigger birds of current market size.

Among various heat stress mitigation strategies, ventilation improvement was preferred by producers. Various studies were conducted to verify the impact of air velocity (AV) on the thermal comfort of broilers under stressful conditions [20,24,25,26]. Furlan [27] suggested that a 50% increase in AV will help maintain optimum weight gain and feed gain ratios for heavy broilers under hot weather. Moreover, they found an increase in AV decreases skin Ts, but those are for birds aged up to 35 d. As AV is found to be potential in mitigating heat stress, it is also important to know the responses of birds under different AVs.

Since Ts is useful in assessing the thermal stress of broiler chickens [18,21], several approaches were taken to build models and methods that can be easily used to practically predict the Ts of broilers [20,23,28]. However, limitations in Ts measurement, changes of Ts with the environment, and variations of Ts at various ages underestimated the Ts of broiler chicken. Moreover, all the previous models were established on the Ts found from broilers of age up to 42 d only. Thus, an updated Ts prediction model which can be used for current market-sized broiler chicken is needed.

It is not only the production house Ta that can determine the comfort alone as the birds’ heat loss depends on the difference between body Ts and Ta [28,29]. Moreover, RH and AV strongly influence the thermal comfort of birds. Thermal indices including temperature and humidity govern the comfortable thermal environment of broilers. Hence, the assessment of the broilers’ Ts at stressful environmental conditions is vital to design a comfortable ambiance for the birds and ensure their performances and welfare. Broiler Ts can be used as an indicator of thermoregulatory status. Moreover, thermal imaging can help assess the condition without touching the birds and also for the birds in commercial production houses [23]. The Ts variation with environmental and physiological factors was investigated through some research, but none of them were conducted for the heavy broilers of age 42–61 d, nor the whole flock of birds. This study was aimed to fill that gap. The experiment was designed to obtain temporal and spatial variations of Ts for different ages and environmental conditions under heat stress. The variation in Ts was investigated under two dynamic AV treatments—high and low. Additionally, a regression model between Ts and environmental factors (AV, T-RH-index) and age was established to help assess the thermal comfort of heavy broilers in commercial flocks under heat-stressed conditions and take necessary management decisions.

## 2. Materials and Methods

### 2.1. Experimental Unit

The experiment was carried out in the poultry engineering laboratory (PEL) of NCSU for three consecutive summers from 2017 to 2019. In 2017 and 2018, two flocks of birds were experimented each summer, but only one flock was studied in 2019. Hence, heat stress experiments were conducted on a total of five flocks of heavy broilers.

The PEL has six simulated poultry chamber systems with the core chambers in 2.44 m × 2.44 m × 2.44 m dimension. All these chambers are equipped with a nipple drinker line, four feeders, and an automated switch-timer soft lighting system (Figure 1). Each of the chambers was comprised with a blower house, a conditioning chamber, a turnaround, and an exhaust duct. A belt-driven blower controlled by a variable frequency drive (VFD) system provides various ventilation rates and desired airspeeds in the range of 0.9–4.6 m/s at birds’ height according to birds’ age and ambient condition. In each of the blower houses, at the outlet of the blower, an adjustable damper controls the amount of fresh air entering the system based on the chamber inlet temperature and blower’s revolution per minute (RPM). More detailed descriptions of the PEL and operations are reported by Wang-Li [30].

### 2.2. Animals

Before each flock, a total of 400 male broilers (“Ross-708” for flocks 1–4, “COBB 500” for flock 5) were hatched and raised in the floor pans under similar conditions at the NCSU poultry unit. Then, 264 birds without leg defects were selected randomly to be placed in the six experimental chambers with 44 birds per chamber at the age of 28 d. They remained in the chamber until the age of 61 d. The final stocking density was ≤40 kg/m^2^, followed by the animal welfare guideline.

### 2.3. Core Chamber Environmental Data Monitoring

In each core chamber where birds were housed, the Ta and RH were monitored continuously with a Thermocouple, and HOBO Pro v2 External T/RH Data Logger, Model U23-002 (Onset, Computer Corporation, MA, USA) placed at the airflow inlet and outlet of the chamber at birds’ height. Calibrated thermocouples (range: −5 °C to 50 °C, and accuracy: ±0.002 °C) recorded temperature data continuously at 1 min interval. All the thermocouple measurements were integrated into 6 PLC control boards for data acquisition onsite readings. The HOBO sensors recorded both temperature and RH at 10 min intervals. Air velocities were tested and pre-determined under different blower frequencies in Hz displayed in the VFD system.

### 2.4. Air Velocity Treatments

Increasing AV is an effective way to improve broiler performance and welfare. However, there is no fixed range or value of AV established for heavy broilers yet since the AV requirements depend on weather conditions. Birds’ body weight and age also impact the ventilation requirement. There are some guidelines about AV that can bring wind-chill effect on effective temperature [31,32]. On top of that, Czarick and Fairchild [33] reported that for heavier birds, air velocities 50% higher than usual (up to 3 m/s) would help maintain optimum weight gain and feed gain ratio during hot weather. Additionally, Yahav [26] found that the AV of 1.5 to 2.0 m/s is optimal to maintain the performance of broilers under harsh summer conditions. Two sets of dynamic AV treatments (high AV and low AV) were designed to test the broilers’ response to AV for this study. The treatment velocities were designed by the research team based upon birds’ possible responses to air temperature classes and birds’ age. Table 1 and Table 2 show the high and low AV treatment designs. The difference between the two treatments varied in range depending on birds’ age and how far the measured chamber temperature differed from the optimal thermal condition. It is important to note that compared to the previous broiler studies [25,26,34] with static AV treatments, this study implements dynamic AV treatment levels based on the age of the birds and temperature of the air around the birds, which is highly recommended and widely used in the broiler industry.

The AV treatment started on the birds at the age of 35. As shown in Table 1 and Table 2, when the Ta was below optimal temperature, there was no AV differences between the treatments to avoid cold stress to complicate the experiments. From 35–61 d of age, high AV treatment was applied to chambers 1, 3, 5 and the low AV treatment on chambers 2, 4, 6. Change in AV was achieved by changing variable speed blowers’ frequency of the VFD.

### 2.5. Thermal Image Capturing

Infrared thermography has been widely used to estimate the surface temperature of chickens [18,19,35,36,37]. Birds’ head surface temperature was considered surface temperature (Ts) as it was easier to get the skin temperature from the head than the other body parts with many feathers. Top-view thermal images of birds were captured using handheld infrared cameras, FLIR T400 (Teledyne FLIR LLC, Wilsonville, OR, USA) for flocks 1–4 and FLIR E8 (Teledyne FLIR LLC, Wilsonville, OR, USA) for flock 5. The opening of the door was kept minimal so that the birds were assumed undisturbed during image capture. For every flock, images were captured on randomly selected days between 42–60 d at different times of the day and at different ages. Each core chamber was virtually divided into the following six segments (see Figure 1 and Figure 2 for relative locations): 3 in the outlet area (feeder-1, back-middle, feeder 2); 3 in the inlet area (feeder-3, front-middle, feeder-4). Six images (one image per segment) were taken in each core chamber during every image taking time at early morning (5:00–7:00), morning (7:00–10:00), late morning (10:00–12:00), noon (12:00), early afternoon (12:00–14:00), late afternoon (14:00–16:00), evening (17:00–19:00), and night (19:00–22:00), respectively. Figure 2 illustrates representative images of six segments in one chamber. Images were downloaded by “FLIR” software (Teledyne FLIR LLC, Wilsonville, OR, USA), which allowed reading temperatures in °C. All six images’ total countable head temperature were averaged and considered as birds’ surface temperature in the given chamber.

### 2.6. Temperature Humidity Index

Since there is no equation yet established to calculate heavy broilers’ temperature humidity index (THI) directly from Ta and RH, this study used the following equation established by [38]:THI = 0.8 × Ta + RH (Ta − 14.3)/100 + 46.3(1)

Ta = air dry-bulb temperature (°C)

RH = relative humidity of air (%)

The calculated indices were then used to identify the thermal comfort for the broilers, according to Table 3. The thermal environments were classified into five comfort/discomfort conditions (Table 3) following the method by [38]. The classification was adopted by Moraes [38] from the combination of average values of temperature, and relative humidity recommended to commercial broilers and laying chickens by several other researchers mentioned in the literature, hence we assumed this could be applied in the current study.

### 2.7. Data Analysis

Each of the five flock experiments was conducted in a completely randomized experimental design with two treatments (high AV and low AV) and in subplots with three replications for each treatment. So, while checking the difference between two treatments within a flock, we used Student’s t-test when there was only one factor. The number of samples used was *n* = 3 since we had three replicates under each treatment within a flock. While checking the differences for multiple factors, the data were evaluated through analysis of covariance (ANCOVA) and the means compared using the Tukey test at the level of 5% probability. When investigating the significant difference for the high and low treatment for the overall flocks, the total samples were 15 for each treatment. The descriptive analysis such as mean, median, quantile, standard deviation of any parameter was performed using the statistical software RStudio (version 1.0.143) (RStudio, Boston, MA, USA). Statistical analysis was also conducted with the same software.

## 3. Results

### 3.1. Environmental Conditions

All the five flocks’ experiments were conducted under summer conditions. Table 4 represents the average environmental conditions at the core chamber inlets while the thermal images were captured. There were no significant differences in Ta, RH, and THI between treatments in any flock. The AV treatments were designed to be dynamic, i.e., changed according to the inlet Ta and age of the birds (Table 1 and Table 2). The AV during data collection time was significantly different under two treatments (*p* < 0.05) in each flock (Table 4). The AV under high treatment was always higher than those of low.

The experiment was designed to obtain Ts variation under heat stress conditions. The thermal index was “severe discomfort” on average during the first flock. The rest of the flocks were “moderate discomfort”. Table 5 represents the percentage of time the birds were under heat stress during the experiment. The data presented in Table 5 reflects only the time of image capture, which is around 1–3% of the overall experiment period for each flock. During the image capture time, the chamber environment (inlet Ta and RH) exceeded the optimal growth condition 98.6%, 47.9%, 55.6%, 50%, and 71.7% time of flocks 1 through 5 consecutively (Table 5). Flocks 1 and 5 were more stressed than flocks 2, 3, and 4.

### 3.2. Average Surface Temperature Variation in Flocks

In investigation of the Ts variation under AV treatment over the experiment, ANCOVA test was conducted considering AV Treatment and Flock as main factors (Table 6). It was observed that both AV and Flock significantly impacted (*p* < 0.05) the birds’ average Ts. The differences among flocks (Table 7) were found by Tukey’s HSD test.

The average Ts for heavy broilers from all five flocks is summarized in Table 7. Flocks 1 and 5 had significantly higher Ts (*p* < 0.05) than the other flocks (Table 7). Flocks 2 and 4 observed the lowest average Ts among all the flocks. Mean Ts in the third flock was higher than that of the second flock. The first and fifth flock observed more stressful times than the other three flocks and, hence higher mean Ts for those flocks were reasonable.

### 3.3. Ts Variation with Ta

The variation of Ts with Ta was investigated by building a simple linear relationship between them. It was discovered that the Ts had a positive co-relationship with Ta (Figure 3). This relationship was significant at a significance level of 0.05. Although there was no interaction effect for Ta and AV treatment at α = 0.05 level, Ts was found lower for high AV treatment than that of the low AV treatment for all flocks. Besides Ta and AV, age was also found to impact Ts for these heavy birds significantly.

### 3.4. Surface Temperature Variation with Temperature Humidity Index

Like the Ta, the broilers’ thermal comfort depends on RH as birds’ respiration also works as a pathway to lose heat under heat stress. Hence, the variation of Ts was checked under various indices for all flocks. Figure 4 indicates that the Ts vary over different THI classes. The THI significantly changed the Ts in any flock. The Ts primarily increase when the THI condition changes from comfortable to life-threatening. The high AV treatment had a significant positive effect on Ts during the second, third, and fourth flocks. Interaction between THI and AV did not change the Ts during any flock.

### 3.5. Surface Temperature Variation over the Time of the Day

The time of the day was divided into six periods. Although the data represents all six time periods, not all flocks had each interval’s representative data. According to Figure 5, the Ts of broilers was highest in the afternoon (14:00–18:00) at any flock and lowest in the early morning (5:00–8:00). ANOVA analysis suggested time of the day significantly impacted (*p* < 0.05) the variation in Ts. Additionally, AV treatments had a significant effect on Ts at any time; the Ts under high AV was consistently lower than that under low AV (Figure 5). The interaction between AV and daytime had no significant effect on Ts variation in any flock.

### 3.6. Ts Variation with Age

The Ts variation of broilers aged from 6th to 9th weeks under both AV treatment for all flocks is presented in Table 8. Under high AV, the Ts decreased significantly (*p* < 0.05) over the week during 2nd and 4th flocks. Under low AV treatments, Ts changes with age during 4th flock. There was no significant difference in Ts under low AV treatment at any week.

A two-way ANCOVA test was conducted to test the effects of age (in week) and the two AV treatments on the Ts of chicken. The main effect of age in week revealed a significant (*p* < 0.05) impact on the broilers’ Ts (Table 9). A significant (*p* < 0.05) positive correlation between Ts and age for the overall flocks was found. The AV treatment also significantly affected the broilers’ Ts. However, the interaction between age and AV treatment was not significant (*p* < 0.05). Although we considered flock as a blocking factor, the ANCOVA test revealed it had a significant impact on the variation of the broilers’ AV. According to Table 4, there were no statistical differences observed in the environmental condition of five flocks. Hence, we considered flock as a blocking factor, and the effects due to flock were not discussed later on.

### 3.7. Surface Temperature Variation at Inlets and Outlets

Table 10 reflects the differences in Ts at the inlets and outlets of the chambers in all flocks. Under high AV treatments, the average Ts at the outlet was not significantly different from that at the inlet for all flocks. This was the same for low AV treatment. Air entered the chamber through the inlet, and so the birds under heat stress wanted to stay and hang around more at that side. The entered air helped birds cool down, so the Ts were lower at this side. On the other hand, air exited through the outlet. The heat released by the birds exited the atmosphere through the outlet, so the air at the outlet was warmer than the inlet. Hence, the birds experienced higher surface temperature at the outlet.

### 3.8. Regression Modeling

Since the results indicated the heavy broilers’ Ts were affected by age and several environmental factors (Ta, RH, THI, AV), a regression modeling was conducted to predict Ts from the multiple factors. A correlation matrix was first built and analyzed to identify which factors had a significant impact on Ts.

The broilers’ Ts had a strong positive correlation with Ta (0.76), THI (0.68), and AV (0.43) (Figure 6). On the other hand, Ts was negatively correlated with RH (0.71). The Ts was positively correlated with age, although the correlation factor (0.06) was not high. Hence, a regression model was built to predict Ts from all these factors (Ta, RH, THI, AV, and Age). Primarily, a simple linear regression model was built. Then, two-way interaction between all the predictor variables was incorporated into the first model. However, a large variance inflation factor (VIF) was found in both models. The THI is strongly correlated to Ta and RH, so the model had a large variance inflation factor. Hence, the issue was resolved by excluding the model’s correlated factors Ta and RH. Since THI was calculated from Ta and RH, hence the thermal stress condition for chicken can be explained by THI solely.

The proposed model to predict Ts is given below:Ts = −4.40 + 0.11×Age + 0.45 × THI − 0.29 × AV(2)
where, Ts = Surface temperature of broiler (°C)

Age = Age of the chicken (day)

THI = Temperature humidity index

AV = Air velocity (m/s)

According to the model (2), the Ts increases with age and THI. Additionally, Ts is inversely related to AV, i.e., if AV decreases then Ts increase. This model is a significant model at significance value 0.05. The R^2^ value of the above-mentioned model is 0.512.

## 4. Discussion

As heat stress is one of the biggest hindrances in poultry production, mitigation of heat stress is a must-needed step to ensure well-performing birds. Consumers are not only concerned with meat, but the animal welfare is also of a great concern. Both performance and welfare depend on a suitable environment to live in throughout the growth period of birds. Understanding the bird’s response to environmental changes is a necessary action as the environment cannot solely tell the birds’ comfort. The birds’ body adjusts to the environment through different changes in extreme environmental conditions. So, it is crucial to recognize the birds’ responses to environmental changes besides the sensors put in place.

Through a set of live broiler heat stress experiments under summer conditions, it was discovered that the average Ts of birds aged 41–62 d was 35.89 ± 2.37 °C regardless of the flock and AV treatment. This study indicated the effect of age on the broilers’ Ts. Cangar [18] observed a decrease in mean Ts from 1st to 6th week of birds. During 2nd and 4th flock, high AV caused a significant decrease in Ts with age. Although it was hypothesized that the Ts increases with age under heat stress, more investigation is required to establish this hypothesis.

Furlan [27], Cangar [18], and Naas [39] measured head Ts of birds under a wide range of Ta (23–32 °C) up to 42 d, and the measured average value were between 27–36.2 °C. This study found that the mean head Ts of the birds from age 42–61 d was 35.89 ± 2.37 °C under heat-stressed conditions. Since the birds were not able to release excess body heat under heat-stressed conditions due to less activity for bigger bodies, the mean Ts was found to be higher than those from the cited studies. While measuring Ts with thermal imaging, measuring from the different body parts is critical. Hence, the head surface, which is exposed to the environment at a bigger age, can be used as a representative area to measure heavier birds’ Ts.

Heavy broilers’ Ts was found to be varied over the changes in Ta. A positive relationship between mean Ts and Ta for young birds (1–7 d of age) under three controlled Ta of 20, 25, and 35 °C treatments was found by Malherois [20]. Nascimento [35] also found the mean Ts increased with the Ta but independent of age. These studies did not verify the Ta effect on Ts for bigger birds of age more than 42 d. In accordance with these studies, current study found the Ts can be impacted by Ta for broilers beyond 42 d. Moreover, age also influences Ts. Since the birds become heavier and their activity decreases with the body weight gain under stressful condition, their heat dissipation rate also lessen. As a result, body Ts increases at a later age.

Air temperature Ta is not only indicative of the broilers’ thermal comfort, but the assessment also needs to be conducted under THI. At any age, the birds’ Ts increases when the THI goes beyond optimal environmental condition, Ts are higher under high THI values. It was also observed that birds were panting more under high THI stressed conditions (Table 3), producing a different vocal signal, indicating that birds need more attention during the higher THI times. Under discomfort to life-threatening thermal conditions, bigger weight of birds made them less active and hence could not actively release excessive heat. This leads to stress and death ultimately.

The diurnal variation of Ts was observed in every flock under heat-stressed conditions. In the morning, the birds remain more active than in the afternoon However, as the Ta rose during the daytime, the birds suffered more from heat stress and became less active, releasing less heat from their body surface. So, higher Ts were observed during noon to afternoon than in the morning period. Additionally, during the experimental period, every morning from 7:00 to 9:00 the screens at the inlet and outlet of each chamber were cleaned, which pushed the birds to move around more and helped release more heat from their body surfaces.

The proposed linear regression model from this study can be used to predict birds’ Ts under heat stress by using only age, THI and AV. Moghbeli [40] also found a mean Ts increasing pattern with age up to 6 weeks due to the feathering index of the head surface. In agreement with Moghbeli [40], the current Ts model indicates that even after 6 weeks of age, the birds’ Ts may increase with age. Under the heat stress conditions, birds became less active and hence were could not release heat from the body surface, so their Ts might increase. Nascimento [23] also established a regression model to predict Ts from feathered and unfeathered areas, but that model was only applicable for the birds of only 42 d. The proposed Ts model upgraded the prediction level for birds of more than 6 weeks. Therefore, it can be used to predict the Ts of heavy broilers. This model developed a positive relationship with THI; hence, the birds’ Ts increases with THI. When the environmental condition exceeds the birds’ thermal comfort zone, it tends to increase the body temperature. Moreover, beyond the thermal comfort zone, birds are no more capable of releasing their metabolic energy as the higher THI does not allow adequate heat transfer anymore. Hence, their body temperature keeps rising. Birds will suffer more at this point. They cannot perform well, i.e., their body weight gain, water intake, and feed intake decline more than usual. The proposed Ts model suggests that the birds’ Ts change inversely with AV. The increase in AV will decrease the Ts of the broiler. Although the time of day had a significant impact on Ts, it was not included in the model as all the thermal image data were captured during the lightning period.

## 5. Conclusions

Broiler surface temperature can be used to indicate their thermoregulatory status under heat stress conditions. Surface temperature varies according to the birds’ age and environmental factors such as Ta, THI, and AV. Through five flocks (35–61 d) of live broiler experiments under summer conditions and two AV treatments, it was discovered that the broiler’s Ts changed temporally and spatially. The variation in Ts happened due to Ta, THI, AV, birds’ age, daytime, and position of birds inside the experimental chambers. The Ts were found lower under high AV treatment. During afternoon time, the broilers’ Ts were higher than other times of the day. It was also found that Ts increased with age and THI. Based upon the experimental data of five flocks, a simple linear regression model was developed to predict Ts from the birds’ age, THI, and AV. Thermal imaging can be used to easily detect surface temperature of broilers under commercial settings. It will help assess heavy broilers’ thermal comfort under heat stress. Producers can take the necessary steps to mitigate heat stress impact by observing surface temperature changes at different ages of birds and environmental conditions.

## Figures and Tables

**Figure 1 animals-12-00328-f001:**
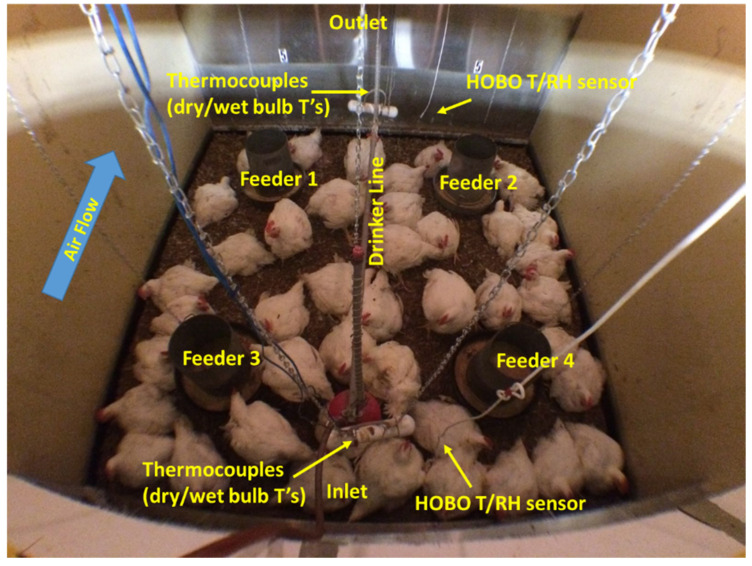
Broilers, feeders, drinker-line, and sensors in the core chambers.

**Figure 2 animals-12-00328-f002:**
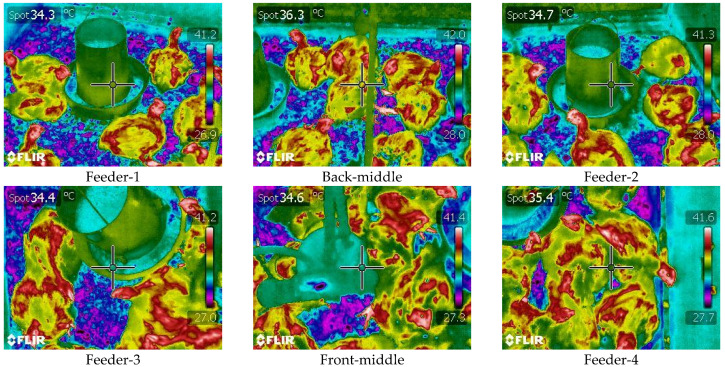
Thermal images by segments in a chamber. Feeder-1, back-middle and feeder-2 were captured in the outlet of the chamber and feeder-3, front-middle and feeder-4 were captured from the inlet of the chamber (see Figure 1 for feeder’s relative locations).

**Figure 3 animals-12-00328-f003:**
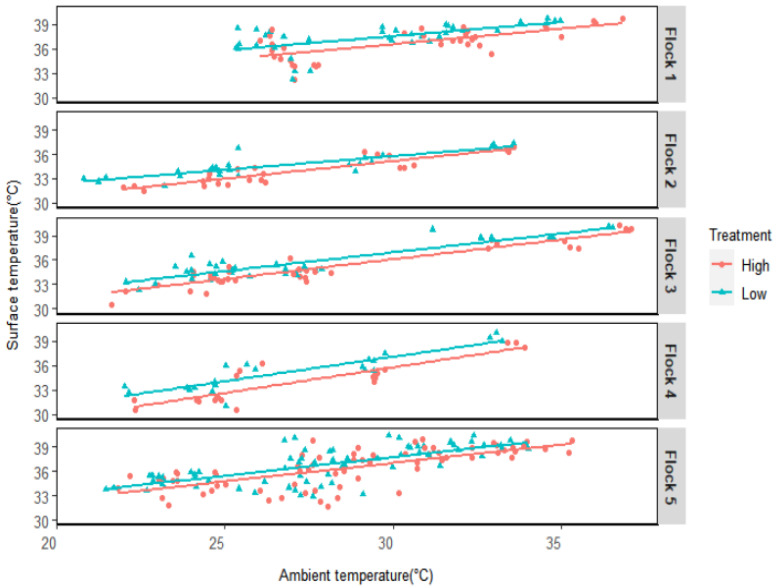
Changes in heavy broilers’ surface temperature with environmental temperature.

**Figure 4 animals-12-00328-f004:**
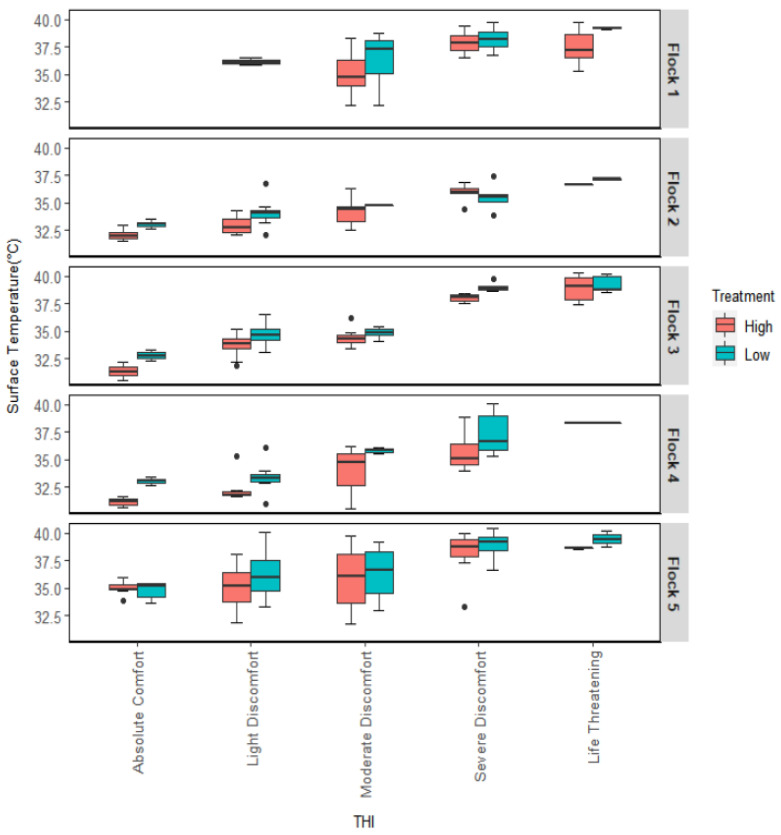
Variation in Ts for different THIs in different flocks under two AV treatments.

**Figure 5 animals-12-00328-f005:**
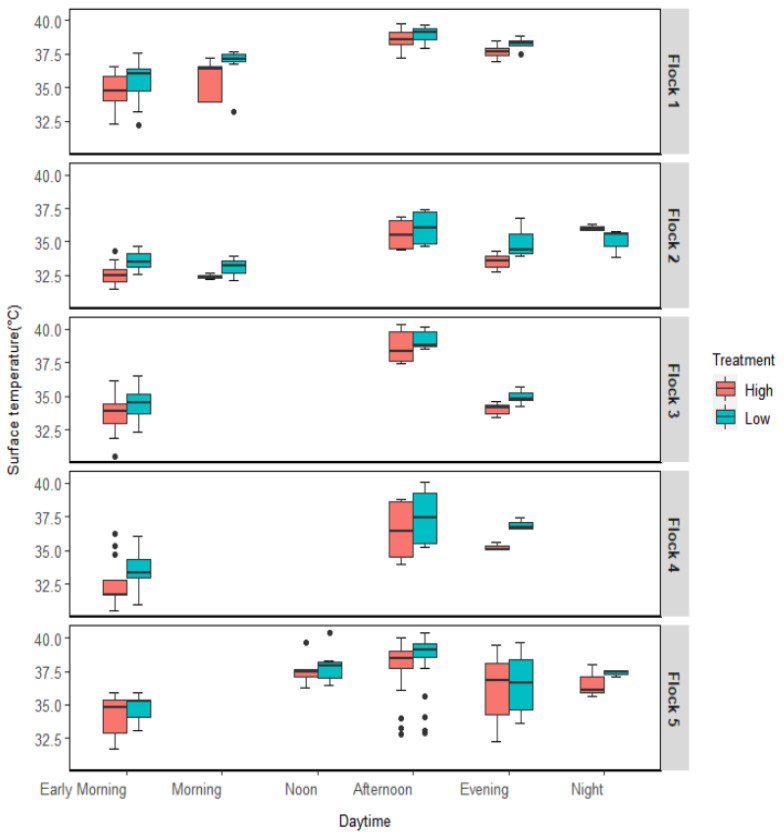
Diurnal variation of heavy broilers’ surface temperature variation under two AV treatments.

**Figure 6 animals-12-00328-f006:**
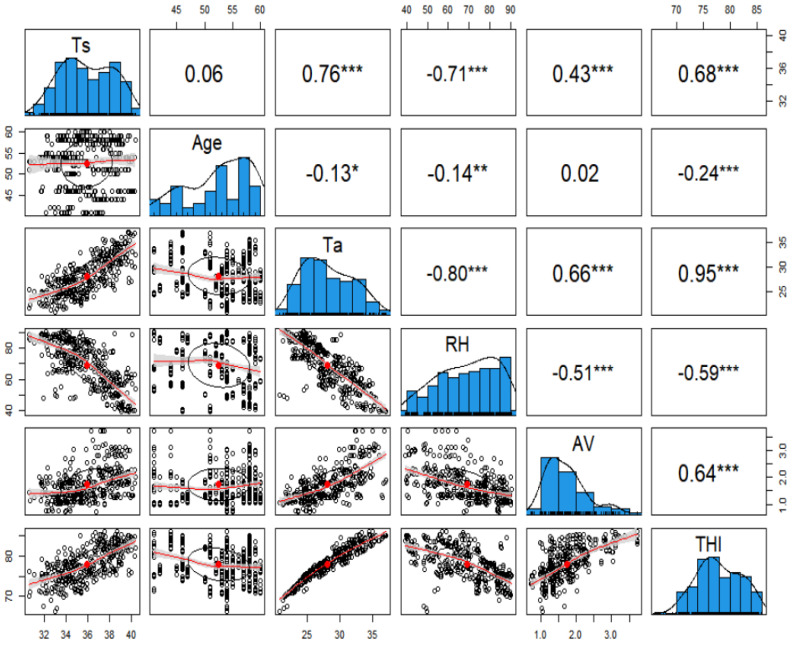
Correlation matrix for the dependent and independent factors. The asterisks (“***”, “**”, “*”) indicate the significance levels of the correlations at different significance level (0.001, 0.01, 0.05).

**Table 1 animals-12-00328-t001:** High AV treatment design.

Broiler Age (Days)	AV (m/s) below Optimum T	Temp °C (°F)	AV (m/s) around Optimum T	Temp °C (°F)	AV (m/s) above Optimum T (Moderate)	Temp °C (°F)	AV (m/s) above Optimum T (Severe)	Temp °C (°F)	AV (m/s) above Optimum T (Life Threatening)	Temp °C (°F)	AV (m/s) above Optimum T (Warning)	Temp °C (°F)
28–34	0.90	<26.0 (78.8)	1.23	26.0–27.8 (78.8–82)	1.33	27.8–28.9 (82–84)	1.48	28.9–32.2 (84–90)	1.64	32.2–33.3 (90–93)	1.75	>37.8 (93)
35–40	0.90	<22.2 (71.0)	1.23	22.0–26.0 (72–78)	2.02	26.0–30.0 (78–86)	2.77	30.0–33.0 (86–92)	3.45	33.0–37.8 (92–100)	3.95	>37.8 (100)
41–42	1.48	<21.1 (70.0)	1.48	22.0–26.0 (72–78)	2.02	26.0–30.0 (78–86)	2.77	30.0–33.0 (86–92)	3.45	33.0–37.8 (92–100)	3.95	>37.8 (100)
43–52	1.48	<20.6 (69.0)	1.75	22.0–26.0 (72–78)	2.02	26.0–30.0 (78–86)	2.77	30.0–33.0 (86–92)	3.95	33.0–37.2 (92–99)	4.33	>37.2 (99)
53–54	1.48	<19.4 (67.0)	1.75	21.1–25.0 (70–77)	2.43	25.0–29.4 (77–85)	3.02	29.4–32.7 (85–91)	3.95	32.7–35.6 (91–97)	4.33	>36.1 (97)
55–56	1.48	<19.4 (67.0)	1.75	21.1–25.0 (70–77)	2.43	25.0–29.4 (77–85)	3.02	29.4–32.7 (85–91)	3.95	32.7–35.6 (91–96)	4.33	>35.6 (96)
57–58	1.48	<18.9 (66.0)	1.75	20.6–25.0 (69–77)	2.43	25.0–29.4 (77–85)	3.02	29.4–32.2 (85–90)	3.95	32.2–35.6 (90–96)	4.33	>35.6 (95)
59–60	1.48	<18.9 (66.0)	2.43	20.6–24.4 (69–76)	3.02	24.4–28.9 (76–84)	3.45	28.9–31.7 (84–89)	4.33	31.7–35 (89–95)	4.43	>35.0 (95)
61	1.48	<18.3 (65.0)	2.43	20.0–23.9 (68–75)	3.02	24.4–28.9 (76–84)	3.45	28.9–31.7 (84–89)	4.33	31.1–33.9 (88–93)	4.60	>33.9 (93)

**Table 2 animals-12-00328-t002:** Low AV treatment design.

AV (m/s) below Optimum T	Temp °C (°F)	AV (m/s) around Optimum T	Temp °C (°F)	AV (m/s) above Optimum T (Moderate)	Temp °C (°F)	AV (m/s) above Optimum T (Severe)	Temp °C (°F)	AV (m/s) above Optimum T (Life Threatening)	Temp °C (°F)	AV (m/s) above Optimum T (Warning)	Temp °C (°F)
0.90	<26.0 (78.8)	1.23	26.0–27.8 (78.8–82)	1.33	27.8–28.9 (82–84)	1.48	28.9–32.2 (84–90)	1.64	32.2–33.3 (90–93)	1.75	>37.8 (93)
0.90	<22.2 (71.0)	1.23	22.0–26.0 (72–78)	1.48	26.0–30.0 (78–86)	2.02	30.0–33.0 (86–92)	2.77	33.0–37.8 (92–100)	3.45	>37.8 (100)
1.48	<21.1 (70.0)	1.48	22.0–26.0 (72–78)	1.48	26.0–30.0 (78–86)	2.02	30.0–33.0 (86–92)	2.77	33.0–37.8 (92–100)	3.45	>37.8 (100)
1.48	<20.6 (69.0)	1.48	22.0–26.0 (72–78)	1.75	26.0–30.0 (78–86)	2.43	30.0–33.0 (86–92)	3.02	33.0–37.2 (92–99)	3.65	>37.2 (99)
1.48	<19.4 (67.0)	1.48	21.1–25.0 (70–77)	1.75	25.0–29.4 (77–85)	2.43	29.4–32.7 (85–91)	3.02	32.7–35.6 (91–97)	3.65	>36.1 (97)
1.48	<19.4 (67.0)	1.48	21.1–25.0 (70–77)	1.75	25.0–29.4 (77–85)	2.43	29.4–32.7 (85–91)	3.02	32.7–35.6 (91–96)	3.65	>35.6 (96)
1.48	<18.9 (66.0)	1.48	20.6–25.0 (69–77)	1.75	25.0–29.4 (77–85)	2.43	29.4–32.2 (85–90)	3.02	32.2–35.6 (90–96)	3.65	>35.6 (95)
1.48	<18.9 (66.0)	1.75	20.6–24.4 (69–76)	2.43	24.4–28.9 (76–84)	2.77	28.9–31.7 (84–89)	3.65	31.7–35 (89–95)	3.80	>35.0 (95)
1.48	<18.3 (65.0)	1.75	20.0–23.9 (68–75)	2.43	24.4–28.9 (76–84)	2.77	28.9–31.7 (84–89)	3.65	31.1–33.9 (88–93)	3.95	>33.9 (93)

**Table 3 animals-12-00328-t003:** Broiler comfort levels under different THI.

THI	Birds Comfort
≤72	Absolute comfort
73–76	Light discomfort
77–80	Moderate discomfort
81–84	Severe discomfort
≥85	Life-threatening

**Table 4 animals-12-00328-t004:** Average environmental conditions in chambers during the image data collection times.

Flock	AV Treatment	Ta (°C)	RH (%)	THI	AV **
1	High	30.38 ± 3.35	67.86 ± 11.67	81.17 ± 3.25	2.51 ± 0.65 ^a^
	Low	29.54 ± 3.09	69.14 ± 12.10	80.14 ± 2.95	1.68 ± 0.51 ^b^
2	High	27.09 ± 3.54	72.62 ± 18.85	77.01 ± 4.48	1.93 ± 0.85 ^a^
	Low	26.37 ± 3.68	73.85 ± 14.12	76.20 ± 5.20	1.22 ± 0.46 ^b^
3	High	27.75 ± 4.61	72.09 ± 16.15	77.49 ± 4.30	1.88 ± 0.44 ^a^
	Low	27.39 ± 4.23	72.79 ± 15.06	77.15 ± 4.09	1.43 ± 0.34 ^b^
4	High	27.26 ± 3.59	77.98 ± 11.52	77.84 ± 4.02	1.61 ± 0.26 ^a^
	Low	27.01 ± 3.52	78.32 ± 11.08	77.50 ± 4.04	1.25 ± 0.09 ^b^
5	High	28.56 ± 3.63	62.64 ± 12.71	77.68 ± 3.79	1.99 ± 0.53 ^a^
	Low	27.93 ± 3.43	63.62 ± 12.85	76.95 ± 3.68	1.43 ± 0.37 ^b^

Means within flocks with different letter superscripts are significantly different at (*p* < 0.05); ** the AV during the image collection time.

**Table 5 animals-12-00328-t005:** Distribution of AV treatments implemented from 35 d to 61 d during the image-capturing time.

% of Occurrences						
Flock	AV below Optimal	AV around Optimal	AV Moderate	AV Severe	AV Life-Threatening	AV Warning
1	0.0	1.4	48.6	30.6	19.4	0.0
2	12.5	39.6	29.2	8.3	10.4	0.0
3	0.0	44.4	30.6	8.3	16.7	0.0
4	0.0	50.0	26.2	11.9	11.9	0.0
5	0.0	28.3	26.8	24.6	16.7	3.6

**Table 6 animals-12-00328-t006:** Results of analysis of covariance test for differences in surface temperature under AV treatment.

Source	Degrees of Freedom	Type III Sum of Squares	Mean Square	F-Value	*p* > F
AV Treatment	1	4.08	4.082	40.05	1.43 × 10^−6^ ***
Flock	4	34.94	8.74	85.69	7.53 × 10^−14^ ***

*** Numbers with asterisk represent the significant effects (confidence interval of 95%).

**Table 7 animals-12-00328-t007:** Mean surface temperatures of heavy broilers under two AV treatments in the summer condition.

Flock	AV Treatment	Average Ts (°C)	
1	High	36.68 ± 1.86	37.02 ± 1.84 ^A^
	Low	37.36 ± 1.78	
2	High	33.89 ± 1.69	34.21 ± 1.60 ^C^
	Low	34.53 ± 1.47	
3	High	34.98 ± 2.39	35.35 ± 2.29 ^B^
	Low	35.72 ± 2.17	
4	High	34.04 ± 2.59	34.64 ± 2.54 ^BC^
	Low	35.26 ± 2.34	
5	High	36.35 ± 2.34	36.56 ± 2.27 ^A^
	Low	36.76 ± 2.21	

^A–C^ Means followed by the different letter within flock differ significantly (*p* < 0.05).

**Table 8 animals-12-00328-t008:** Heavy broilers’ surface temperatures variation with age.

Flock	Treatment	Surface Temperature °C (Mean ± SD)			
		6th Week	7th Week	8th Week	9th Week
1	High	NA	$37.10\pm 0.60{\;}^{a}$	$37.23\pm 0.35{\;}^{a}$	$35.74\pm 0.61{\;}^{a}$
	Low	NA	$37.71\pm 0.61{\;}^{a}$	$37.84\pm 0.27{\;}^{a}$	$36.53\pm 1.09{\;}^{a}$
2	High	$34.70\pm 0.57{\;}^{a}$	NA	$33.07\pm 0.36{\;}^{b}$	NA
	Low	$35.07\pm 0.57{\;}^{a}$	NA	$33.97\pm 0.71{\;}^{a}$	NA
3	High	NA	$35.35\pm 0.61{\;}^{a}$	$34.69\pm 0.62{\;}^{a}$	$34.90\pm 0.71{\;}^{a}$
	Low	NA	$35.89\pm 0.47{\;}^{a}$	$35.37\pm 0.46{\;}^{a}$	$35.89\pm 0.51{\;}^{a}$
4	High	NA	$35.25\pm 0.35{\;}^{a}$	$32.43\pm 0.39{\;}^{b}$	NA
	Low	NA	$36.37\pm 0.36{\;}^{a}$	$33.77\pm 0.69{\;}^{b}$	NA
5	High	NA	NA	$35.49\pm 0.79{\;}^{a}$	$36.70\pm 0.94{\;}^{a}$
	Low	NA	NA	$35.94\pm 0.68{\;}^{a}$	$37.08\pm 0.61{\;}^{a}$

^a,b^ Means under each treatment within Flocks with different superscripts are different at (*p* < 0.05).

**Table 9 animals-12-00328-t009:** Results of analysis of covariance test for testing age effect on broiler surface temperature.

Source	Degrees of Freedom	Type III Sum of Squares	Mean Square	F-Value	*p* > F
Week	3	112.9	37.62	8.496	1.82 × 10^−5^ ***
AV Treatment	1	39.1	39.14	8.838	0.00315 **
Flock	4	336.3	84.97	18.984	3.54 × 10^−14^ ***
Week × AV Treatment	3	1	0.33	0.075	0.97345

The asterisk (**,***) represent the significance at different level (0.01, 0.001).

**Table 10 animals-12-00328-t010:** Difference in surface temperatures at the inlets and outlets of the chambers.

Flock	AV Treatment	Ts (Mean ± SD)	
		Outlet	Inlet
1	High	36.84 ± 1.82	36.75 ± 2.01
	Low	37.76 ± 1.77	37.22 ± 1.95
2	High	33.97 ± 1.76	33.79 ± 1.83
	Low	34.49 ± 1.49	34.58 ± 1.78
3	High	35.09 ± 2.39	34.86 ± 2.50
	Low	35.82 ± 2.08	35.61 ± 2.32
4	High	34.02 ± 2.76 ^a^	34.06 ± 2.71
	Low	35.56 ± 2.30 ^b^	34.95 ± 2.72
5	High	36.66 ± 2.37	36.04 ± 2.52
	Low	37.02 ± 2.33	36.67 ± 2.25

Different superscript in a column under each flock indicates values were significantly different (*p* < 0.05) in columns.

## Data Availability

Data presented in this study can be available upon request from the corresponding author.

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
