# Peer review of "Impacts of Air Velocity Treatments under Summer Condition: Part I—Heavy Broiler’s Surface Temperature Response"

_animals, 2022, doi:10.3390/ani12030328_

Round 1

Reviewer 1 Report

Table 1 & 2, how are the bins and air velocities in bins determined for high and low AV treatments?

Line 148- 151, were the IR images taken manually or automatically? In other words, were there any disturbance when images were taken?

Line 154, how often were images taken, daily, weekly, or random-selected days?

Line 169, what’s the unit of Ta in the equation?

Table 3, please fix the notation “<-” and “>-”.

Line 176, if there were five flocks in total, why “3 replications”?

Figure 4 caption, in different “weeks” should be “flocks”.

Table 8, it seems that there was no statistical sig. difference of Ts between Outlet and Inlet for the most part. So in Line 263-264, please rephrase it to, “Under both High and Low AV treatments, the average Ts at the outlet were not significantly different with that at the inlet for all flocks.”

Line 265, change “hot birds” to “birds under heat stress”.

Table 7, data are not conclusive regarding the correlation of Ts with age. The possible correlation could be a confounding factor of higher AV in result of older age. Hence, why not test the model without age as independent variable?

Line 290-291, does the prediction of Ts indicate the physiological condition of chickens? What’s the bounding range this equation applied, i.e. what AV range, etc.

Line 310, be careful again about the statement of “a decrease of broiler’s Ts with age except for 5th flock”.  Make sure this is not caused by the higher AV based on age (temperature setpoint). Since your own data showed inconsistency, I don’t think it’s correct to conclude as you did in Line 311-312.

Line 318, what does “is little over this range” mean? Were Furlan, Candar etc. measured under heat stress, or a wide range including thermal neutral?

Line 330, again please refer to data/evidence when making statement regarding Ts vs, age!!

Line 342-343, one could argue that higher Ts is a physiological coping mechanism of chicken in an attempt of losing metabolic heat. It’s not necessarily a bad thing. The authors should focus on how it can be used as a stress indicator quantitatively.

Line 368, how is Ts a better indicator of thermoregulatory status than THI?

How does the study or model developed help to assess heat stress condition?

Conclusion, the authors should make sure that the statements are precise. For example, “The Ts found lower under High AV treatment. (ln 373)” is not completely correct. Ts is general lower in early morning, this time coincide with lower AV. 

Author Response

Respected Reviewer,

Thank you for your comments and suggestions to improve this paper. Responses to each comment are provided in the attached. We hope the edits to the specific comments below aid in understanding. 

Sincerely,

Suraiya

Reviewer 2 Report

Comments on Akter et al.

This paper addresses an important issue and concludes with a useful model for specifying the conditions for avoiding heat stress in older broiler chickens. However, it is quite difficult to follow exactly how the authors carried out their experiment and how they carried out the statistical analysis. This makes it difficult to evaluate the paper as it is currently written. The lack of behavioural data on the birds’ welfare in different environmental conditions also detracts from the papers’ impact. There are some grammatical/language confusions that need to be corrected.

Specific comments
Line 50. Can’t broilers release heat by panting?

Line 68. Does the ‘strong correlation’ between DBT and Ts mean that Ts and Ta can be used interchangeably? Or is there a conversion factor?

Line 89. Similar comment. “Broiler Ts can be used as an indicator of thermoregulatory status”. Exactly what is the evidence for this?

Line 91. “Although several research was done” is not grammatically correct.

Line 110. The use of the 6 chambers was not clear. Were the 6 chambers  completely independent? Were there 6 chambers, 3 of which were used for High AV and 3 of which were used for Low AV at any one time, with 2 replicates in 2017, 2 in 2018 and 1 in 2019? If so this would give 15 Low AV and 15 High AV. However, in line 109 states that there were 5 flocks. There is some confusion about how many independent replicates of each treatment there were.

Line 135. What does “desirable” mean? How is the 2.03 m/s arrived at?

Line 137. How were the High and Low AV treatments chosen?  Why not much higher than  2.03 m/s for High and lower for Low?

Table 1. What are the figures shown? In the data analysis section (2.7) both medians and means are mentioned as are quartiles and sds. Which are shown here?

Table 3. Is there any independent evidence for the bird comfort  categories? For example, do birds start panting or showing any measurable behavioural changes at particular ‘comfort’ levels.

Line 174. 2.7 Data analysis. It would be helpful to have a clearer description of the ANOVA used, degrees of freedom and whether p values (as in Table 4) are one or two tailed.

Line 179.Does ‘quantile’ mean Quartile? Ie InterQuartile range to go with Median?

Line 215. The phrase “in every flock bird’s average” is confusing. Does it mean that for each of the 5 flocks the average Ts of all replicates for High AV was lower than for all replicates for Low AV? How was this conclusion arrived at? It looks as though it was an individual comparison between High AV and Low AV conditions for each flock (averaged over 3 chambers). Why was the comparison done this way? Surely it should come out as a factor in the ANOVA?  But there is no ANOVA table to check this against.

Lines 300-307. The importance of understanding the birds’ responses to environmental change is acknowledged. So why didn’t you measure this? Why didn’t you measure welfare?

Line 310. The 5th flock was Cobb while the others were Ross. Are the breeds possibly different?

Author Response

(The authors gave the same response as above.)

Reviewer 3 Report

The manuscript investigates the variation of the broiler surface temperature considering two different air velocity treatments, namely high air velocity and low air velocity. The surface temperature is analyzed considering various parameters -such as time of the day and age- and a linear regression model was developed to estimate surface temperature from parameters that can be easily measured on field. Similar works are present in literature, but the novelty of this investigation relies on its focus on heavy broilers (i.e., 42-61 d age). Some of the results of this work are trivial since, for example, it is well known that higher air velocities entail lower surface temperatures (lines 214 -217). Nevertheless, the experimental approach is interesting, and the developed regression model can be useful for future investigations. Unfortunately, the manuscript needs strong improvements, especially in style and specific points should be clarified by the Authors. For this reason, the manuscript should be reconsidered after major revisions.

In their revisions, the Authors should address the following points:

  • the style of the manuscript is not suitable for a scientific journal. The use of the first person must be avoided in the manuscript (e.g., lines 194 and 229) and a careful reading of the entire manuscript is suggested to avoid typos, such as in line 370 or in table 1. Furthermore, it would be better to avoid the use of lumped citations (e.g., lines 52 and 55). Please report the contribution of each work that is cited in the manuscript and that is considered essential in the framework of this investigation. Finally, including a nomenclature at the beginning of the manuscript may be very useful for the reader since abbreviations and variables (please, use subscripts) are presented in the text.

  • One of the pillars of this investigation is the difference between the two air velocity treatments, namely High AV and Low AV. Nevertheless, the main differences between these two treatments are not clearly explained in the manuscript. In Tables 1 and 2, data regarding these treatments are provided but they were not properly described in the text. In addition, it is not clear why there is a heat stress indication (e.g., severe and moderate) in some of the columns while the thresholds of THI are defined later in the text (table 3). Please explain.

  • Authors assess the thermal comfort using THI, as reported in Eq.1. This is an approach that is common in literature, but some main points should be addressed by Authors:

-Authors define comfort levels according to the thresholds that are defined in [37] but in [37] a different THI formulation is adopted. Do the thresholds apply also to the formulation adopted in the present manuscript?

-the thresholds presented in [37] come from https://doi.org/10.1590/S1516-635X2010000100003, but, there, such thresholds are not reported. May you check the reliability of this citation or, alternatively, cite another work?

-does the adopted formulation of THI and the considered thresholds apply also for heavy broilers (42- 61 d)? This is since in [37] the analyses were focused on broilers up to 41 days of age. If this is an assumption of the Authors, it should be explicitly stated in the manuscript.

  • One of the most interesting points of this work is the development of a regression model for the estimation of broiler surface temperature. It could be interesting to provide the reader with a short background about this topic, by citing similar regression models present in literature and the main differences with the presented one.

  • In section 2.3, Authors mention the sensors that were used for measuring indoor air temperature and relative humidity, but they do not provide any details about the accuracy of the adopted instruments. Please report accuracy in brackets. Furthermore, Authors mention the commercial product “HOBO”. If a commercial product is mentioned, it could be useful to provide data about manufacturer and the adopted model.

Additional minor revision:

Line 133: HZ? Is it Hertz? If it applies, please use Hz.

Table 1: Please indicate the temperature in °C out of brackets, while the temperature in °F in brackets. There is a typo in cell 2x2.

Table 2: there is a typo in cell 10x8

Line 169: the variables of Eq. 1 are different if compared to the ones in the text.

Line 370: please correct “brioler”

Author Response

(The authors gave the same response as above.)

Round 2

Reviewer 2 Report

The authors seem to have answered most of the queries and the paper is now much clearer.

Author Response

Respected Reviewer,

Thank you for your valuable guidance to improve this paper. We appreciate your time and effort. 

Sincerely,

Suraiya

Reviewer 3 Report

The Authors strongly worked to address the Reviewers' comments. The improvements are evident in the revised version of the manuscript. Thank you very much for your work.

My recommendation is to accept the manuscript after addressing the following minor revisions:

  • First person pronouns and adjectives are still present in the manuscript; please check lines 16, 236, 242, 243, 335, 338, 361, 363, 364, and 372.
  • Line 237: please put the caption of Table 3 in a new line
  • Line 172: the accuracy of the thermocouple is ±0.0020 °C. This value seems very low. Please check it.

“Lumped citations” means to group references such as [2]-[11], as done in line 52. Instead, it would be recommendable to summarize the main contribution of each referenced paper in a separate sentence and by including the reference number. In this way, the actual contribution of the referenced paper can be easily shown to the reader.

Author Response

Respected Reviewer,

Thank you for your valuable guidance to improve this paper. We appreciate your time and effort. Responses to each comment (in bold) are provided below. We hope the edits to the specific comments below aid in understanding.

Sincerely

Suraiya
